# A controlled study to assess the effects of a Fast Track (FT) service delivery model among stable HIV patients in Lusaka Zambia

Carolyn Bolton Moore[1,2]☯, Jake M. Pry[1,3]☯*, Mpande Mukumbwa-Mwenechanya[1], Ingrid Eshun-Wilson[4], Stephanie Topp[5], Chanda Mwamba[1], Monika Roy[6], Hojoon Sohn[7], David W. Dowdy[8], Nancy Padian[8], Charles B. Holmes[9], Elvin H. Geng[5], Izukanji Sikazwe[1]

1 Centre for Infectious Disease Research in Zambia (CIDRZ), Lusaka, Zambia, 2 University of Alabama, School of Medicine, Birmingham, Alabama, United States of America, 3 University of California, School of Medicine, Davis, California, United States of America, 4 Washington University, School of Medicine, St Louis, Missouri, United States of America, 5 James Cook University, College of Public Health, Medical and Vet Sciences, Queensland, Australia, 6 University of California, School of Medicine, San Francisco, California, United States of America, 7 Johns Hopkins University, School of Medicine, Baltimore, Maryland, United States of America, 8 University of California, School of Public Health, Berkeley, California, United States of America, 9 Georgetown University, School of Medicine, Washington, DC, United States of America

☯ These authors contributed equally to this work.
* jake.pry@cidrz.org, jmpry@ucdavis.edu

**Data Availability Statement:** The data used to prepare these analyses are not public and are part

## Abstract

Fast Track models—in which patients coming to facility to pick up medications minimize waiting times through foregoing clinical review and collecting pre-packaged medications—present a potential strategy to reduce the burden of treatment. We examine effects of a Fast Track model (FT) in a real-world clinical HIV treatment program on retention to care comparing two clinics initiating FT care to five similar (in size and health care level), standard of care clinics in Zambia. Within each clinic, we selected a systematic sample of patients meeting FT eligibility to follow prospectively for retention using both electronic medical records as well as targeted chart review. We used a variety of methods including Kaplan Meier (KM) stratified by FT, to compare time to first late pick up, exploring late thresholds at >7, >14 and >28 days, Cox proportional hazards to describe associations between FT and late pick up, and linear mixed effects regression to assess the association of FT with medication possession ratio. A total of 905 participants were enrolled with a median age of 40 years (interquartile range [IQR]: 34–46 years), 67.1% were female, median CD4 count was 499 cells/mm$^3$ (IQR: 354–691), and median time on ART was 5 years (IQR: 3–7). During the one-year follow-up period FT participants had a significantly reduced cumulative incidence of being >7 days late for ART pick-up (0.36, 95% confidence interval [CI]: 0.31–0.41) compared to control participants (0.66; 95% CI: 0.57–0.65). This trend held for >28 days late for ART pick-up appointments, at 23% (95% CI: 18%-28%) among intervention participants and 54% (95% CI: 47%-61%) among control participants. FT models significantly improved timely ART pick up among study participants. The apparent synergistic relationship between refill time and other elements of the FT suggest that FT may enhance the effects of extending visit

of the Zambian national electronic HIV medical record data, and as such, are only potentially available on request and with permission from the Zambian Ministry of Health (suggested contact: info@moh.gov.zm). Access to data is subject to restrictions including, but not limited to de-identification and suppression of small cell sizes.

**Funding:** This study was funded by the Bill & Melinda Gates Foundation (OPP1115306 to CBH) https://www.gatesfoundation.org/ and National Institutes of Health (K24 AI134413 to EHG) https://www.nih.gov/. The funders had no role in study design, data collection and analysis, or decision to publish.

**Competing interests:** The authors have declared that no competing interests exist.

spacing/multi-month scripting alone. ClinicalTrials.gov Identifier: NCT02776254 https://clinicaltrials.gov/ct2/show/NCT02776254.

## Introduction

To reach HIV epidemic control, countries must offer reliable, high-quality, long-term antiretroviral therapy (ART) to persons on treatment while continuing to absorb new patients accessing care and treatment services. Although scale up of supply chains, task shifting, and universal training to create human resources for health have improved the HIV care landscape, the front lines where the health system interfaces with the public, remain highly burdened. By 2017, patient volumes had grown so rapidly that waiting times for clinical services reached 3–4 hours (or longer) per visit in Zambia [1]. For patients already on treatment, negative experiences accessing care, including unacceptable waiting times, emerged as a critical factor in undermining the sustainability of treatment [2]. The standard of care in Zambia at the time of study initiation included a mean one-month ART refill schedule, (though 3-month refills had been recommended for stable patients in alignment with World Health Organization standards since 2014), HIV was dispensed on a first-come first-served schedule through a single, common queue service point with patient volume often in excess of clinic design and staffing [3,4]. Reducing waiting time for HIV services and improving HIV clinic visit experience represent opportunities to optimize epidemic control in resource-limited settings, like Zambia.

Differentiated service delivery (DSD) models, defined as "client-centered approaches that simplify and adapt HIV services across the care and treatment cascade in ways that both serve the needs of PLHIV better and reduce unnecessary burdens on the health system", represent an overarching public health strategy to enhance the capacity to absorb large numbers of patients by reducing the burden of care [5,6]. Although these DSD models are exciting innovations that, in theory, meet public health needs, impact of an expedited service delivery model on pharmacy visits in Zambia has not been well documented—a critical addition given the inclusion of the FT model in Zambia's differentiated service delivery agenda [7]. In Zambia, routine care requires that patients receive a full examination at each clinical visit as well as receive adherence counseling. Previous research in Zambia has estimated a median wait time for HIV clinic visits of 60 minutes (interquartile range: 40–147 minutes) [8]. We assessed a simple model referred to as "Fast Track" (FT) which offers pre-packed ART refills without the need for full clinical evaluation e.g., avoids a triage evaluation, short counseling session and clinical history and examination. Thus, FT facilitates rapid entry and exit from the clinic for patients who are considered clinically stable, thus reducing the waiting times and the opportunity costs of care. Previous evaluations of FT models have shown benefit in sub-Saharan Africa but may be influenced by selection of those doing well [9–11].

To address this, we compared late ART pick-up in facilities adopting a FT model to sites providing routine ART services/ standard of care in urban Zambian clinics. Sites were selected to be similar in size, geography, and resources. We examined the effect of the FT model on incidence of being late ($>7$ days, $>14$ days, $>28$ days, and $>90$ days) for next pharmacy appointment, medication possession ratio, and time to return after missing or being $>28$ days late for next pharmacy appointment. We also explored potential mechanisms of effect by incorporating changes in appointment spacing during this time.

## Methods

### Ethics statement

Ethics approval was obtained from the University of Zambia Biomedical Ethics Committee, the University of California at San Francisco Institutional Review Board, and the University of Alabama at Birmingham Institutional Review Board. Written informed consent was obtained from all participants. Written informed consent was translated into three local languages (English, Nyanja, and Bemba) and obtained from all adults (≥18 years of age) participants and guardians with assent from participants <18 years of age.

### Setting and population

We evaluate the effects of a the fast-track (FT) model on incidence of late antiretroviral drug pick up among stable patients in HIV care by comparing two facilities where FT was being rolled out to five HIV care facilities maintaining standard of care. The two intervention and five control sites were similar in patient population, located in urban areas, and at the same level of the provincial health systems (Fig 1). All study facilities are run by the Zambia Ministry of Health and supported by the Centre for Infectious Disease Research Zambia (CIDRZ). Within each facility, we recruited a convenience sample of individuals meeting the eligibility criteria for FT according to national programmatic criteria (adherent to HIV care >6 months) between 17 March 2016 and 31 August 2016) [7,12].

### Measurements

In all enrolled patients, we obtain demographic, clinical, and pharmacy (primary outcome) dispensation information from the national electronic HIV medical record. The electronic HIV medical record includes all HIV patient care interactions including antiretroviral therapy (ART) date of interaction and next appointment date (based on dispensation volume/interval). Recruitment information including HIV care ID and informed consent were collected electronically using Open Data Kit (ODK) software. In select cases, physical charts were reviewed to adjudicate information about visits that was not clear in the electronic medical record.

### FT intervention

The FT intervention is a streamlined pharmacy ART pick up system aimed at decreasing patient waiting time and overall time spent at the clinic. The intervention consisted of: 1) a

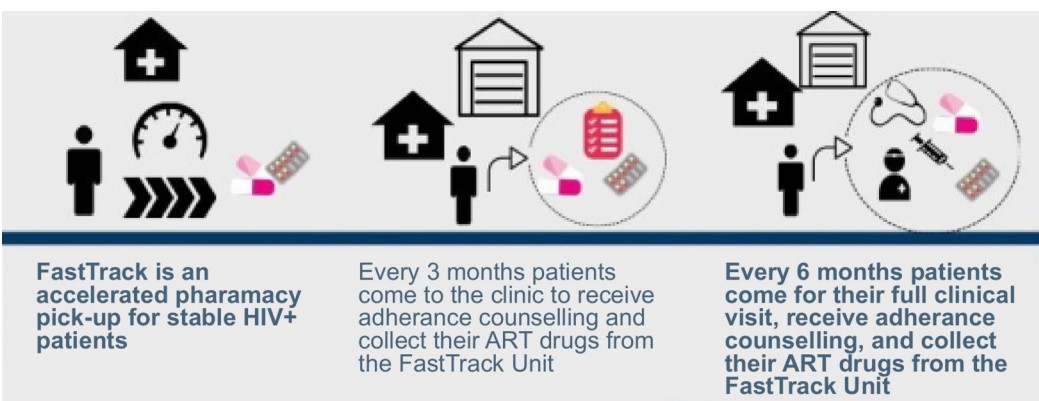

**Fig 1. Infographic describing the fast-track model.**

dedicated FT participant room that was staffed by a pharmacy technologist who pre-packaged and dispensed a 3-month supply of ART, 2) lay health care workers who provided brief symptom screening, adherence counselling, and identified patients that required a higher level of care, and 3) modified clinic encounter schedule including 3-month ART pick-ups where every other visit also included a clinical/medical encounter (every six months, per Zambia MoH guidelines) [8]. At each clinic visit, intervention participants proceeded directly to the dedicated FT room where they received symptom screening, adherence counselling and collected ART drugs. After a clinical encounter was completed, and given that there were no additional medical needs, the intervention participant proceeded to the dedicated FT room where a 3-month pick-up of ART refill occurred.

## Standard of care (control)

The standard of care median visit space during study planning was 1 month ART refill with clinical follow-up visits spaced 3–6 months, but longer appointment intervals (up to three months) began being introduced during the study period. Standard of care queuing operated on a first-come first-served basis with visit appointments on the date, no appointment times. Almost all HIV care visits are completed before 14 hours (2pm) local time. ART clients followed routine clinic processes of triage, adherence counseling, symptom screening with physical examination if required, proceeded to pharmacy to collect ART.

## Outcomes

The primary outcome was the incidence of being >7 days late for a pharmacy appointment. We also assessed the first incident of being >14 days, >28 days, and >90 days late for pharmacy appointment. Additionally, we calculated medication possession ratio (MPR) based of pharmacy refill data and developed a binary variable of at least 90% medication possession or <90% medication possession across the study period. Finally, we assessed time until return after an incidence of being >28 days late for a pharmacy appointment.

## Analytic plan

In our primary analysis, we used Kaplan-Meier failure estimates, stratified by treatment condition, to evaluate the association of FT with the primary outcome of first incidence of being >7 days late for next pharmacy appointment in our one-year follow-up period. Additionally, we used Kaplan-Meier estimates, similarly stratified, to evaluate incidence of late pharmacy pick up at >14, >28, and >90 days late for next pharmacy appointment at one year of follow-up and time to return after an incidence of being >28 days late for next pharmacy appointment during the 60 days of follow-up. Significance was assessed at the 95% confidence level using the log-rank test. We also estimated adjusted Cox proportional hazards to evaluate the effect of FT on being >28 days late adjusting for age, sex, CD4 count, WHO clinical stage, duration of time on ART, marital status, and educational level. We used robust standard errors to account for clustering at the clinic level. As an additional analysis, we used mixed effects logistic regression to estimate the association between FT and high (>90%) medication possession ratio (MPR)—defined as the proportion of time during one year of follow-up which individuals, according to the electronic HIV medical record, had ART.

Finally, in order to explore the hypothesis that FT augmented the effects of longer appointment intervals, we exploited the natural variability in appointment intervals through examining an interaction between pharmacy appointment interval and the FT. To do so, we used a mixed-effects logistic regression with an interaction between FT and appointment interval on greater than 90% MPR. The model adjusted for age, sex, CD4 count, WHO clinical stage,

duration of time on ART, marital status, and educational level with random effects at the individual and facility level. Throughout the analyses, we used multiple imputation to account for missing predictor data (where missing values were found to be <30%) by chaining complete covariate data on sex, age, facility, intervention status, study enroll date, and time in HIV care. Missing pharmacy appointment data in the electronic HIV medical record at study enrollment visit was imputed using median appointment spacing by clinic across the seven control and intervention facilities.

## Sample size considerations

The target sample size at both intervention facilities was 400 patients (200 at each intervention facility) compared to at least 400 control participants enrolled at five control facilities, sufficient to detect an anticipated difference of 40% in incidence of late ART pick up (>7 days) at one year of follow-up time.

## Results

### Patient characteristics

Among 901 patients included in the analysis dataset, median age was 40 years (IQR: 34–46), 607 (67.4%) were women, median CD4 cell count was 499 cells/mm$^3$ (IQR: 354–691), and median time on ART was 5 years (IQR: 3–7) (Fig 2). Intervention and comparison groups differed regarding age, sex, marital status, CD4 cell count at HIV care initiation, and World Health Organization (WHO) HIV clinical stage at HIV care initiation (Table 1). There were fewer males enrolled in the intervention (26.3%) compared to the control (34.2%) group, a greater proportion of more single, divorced, and widowed participants in the intervention group than the control group, a lower median CD4 cell count at HIV care initiation across those in the control group compared to the intervention group, and a greater proportion of intervention participants with a WHO stage > 2 at HIV care enrollment. Intervention and control groups were similar regarding time in HIV care, reported education level and reported household income (Table 1).

### Adherence to scheduled pharmacy visits: Late drug pick-up

During the one-year follow-up period FT participants had a significantly reduced cumulative incidence of being more than 7-days late for ART pick-up (0.36, 95% CI: 0.31–0.41) compared to controls (0.66; 95% CI: 0.57–0.65). This trend extended to an outcome of more than 14 days late for ART pick-up where FT participants had a significantly lower cumulative incidence of 0.32 (95% CI: 0.28–0.37) compared to control participants at 0.61 (95% CI: 0.57–0.65). Fast-track participants had a lower proportion of delayed ART pick-up appointment >28 days beyond their original appointment at 23% (95% CI: 18%-28%) compared to 54% (95% CI: 47%-61%) among control group participants (Fig 3 and S1 Fig) Though not significantly different, a greater proportion of intervention participants returned to care after being 28 days late at 60 days of follow up with 50.5% (95% CI: 41.3, 60.4%) return compared to control participants where 43.1% (95% CI: 36.8, 49.8%) returned (Fig 4). After adjustment, patients in the control facilities had a 2.84 greater chance of missing at least one visit by more than 28 days (95% CI: 1.65–4.89).

### Adherence: Medication possession ratio

We also assessed adherence to HIV care measuring the medication possession ratio for the one-year follow-up period. Using data from pharmacy pick-ups we found that the median

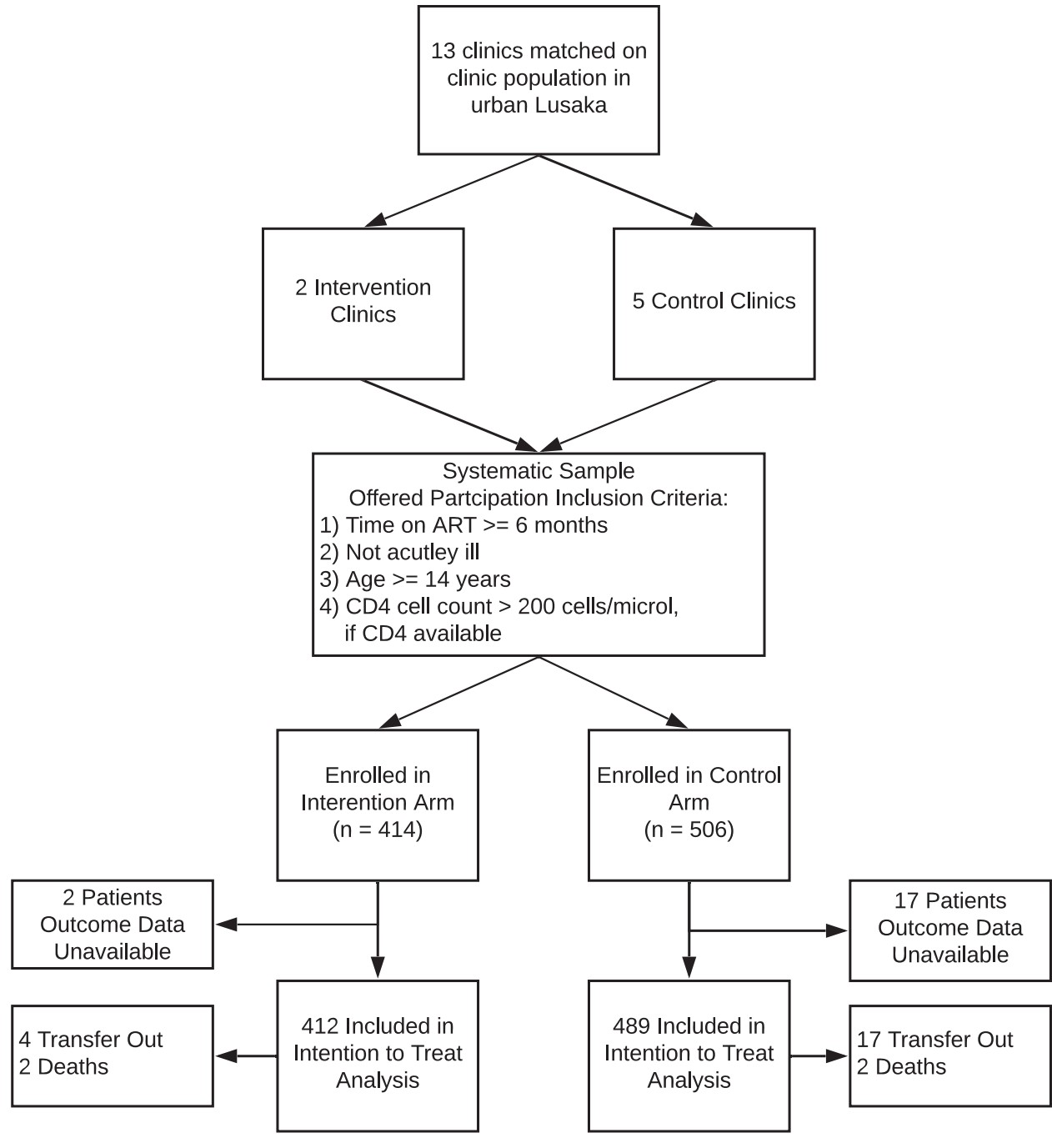

**Fig 2. Flow diagram describing the analysis population.**

medication possession ratio (MPR) for both the intervention and control group was high at 1.0 and 0.99. However, 25% of the control population had an MPR <90% while the intervention group had only 1% below 90% MPR. The adjusted odds ratio for MPR ≥ 90% was significantly higher among those in the intervention group (OR: 3.75, 95% CI: 1.41–9.97) compared to the intervention group (Table 2).

**Table 1. Patient characteristics by intervention status.**

| Factor | Level | Intervention | Control | p-value |
|---|---|---|---|---|
| | | n (%) | n (%) | |
| N | | 412 | 489 | |
| Age | median (IQR) | 34 (33–47) | 40 (33–47) | <0.001 |
| Sex | Female | 305 (74.0%) | 322 (65.8%) | 0.008 |
| | Male | 107 (26.0%) | 167 (34.2%) | |
| Baseline CD4 (cells/mm$^3$) | median (IQR) | 496 (357–640) | 432.5 (294–597) | 0.012 |
| WHO Stage at HIV Care Initiation | 1 | 138 (38.8%) | 225 (51.1%) | <0.001 |
| | 2 | 64 (18.0%) | 95 (21.6%) | |
| | 3 | 139 (39.0%) | 109 (24.8%) | |
| | 4 | 15 (4.2%) | 11 (2.5%) | |
| Years in Care | median (IQR) | 5.8 (2.9–7.9) | 5.5 (3.0–7.6) | 0.275 |
| Marital Status | Single | 48 (11.7%) | 44 (9.0%) | <0.001 |
| | Married | 185 (44.9%) | 261 (53.4%) | |
| | Divorced | 56 (13.6%) | 48 (9.8%) | |
| | Widowed/Widower | 53 (12.9%) | 37 (7.6%) | |
| | Unknown | 70 (17.0%) | 99 (20.2%) | |
| Education Level | None | 11 (2.8%) | 16 (5.9%) | 0.088 |
| | Grades 1–6 | 90 (23.1%) | 52 (19.0%) | |
| | Grades 7–11 | 115 (29.6%) | 83 (30.4%) | |
| | Grade 12 | 100 (25.7%) | 83 (30.4%) | |
| | College/University | 73 (18.8%) | 39 (14.3%) | |
| Household Income (daily) | <5 USD | 227 (98.7%) | 185 (98.4%) | 0.970 |
| | ≥5 USD | 2 (0.9%) | 3 (1.6%) | |

Note: p-value for continuous variables based on t-test and categorical variables based on Chi-Squared test; IQR—interquartile range; USD—U.S. dollar.

There was a significant difference in probability of attending next appointment on time (± 3 days from next appointed visit date) by intervention status despite receipt of extended refill (>85 days) (Fig 5).

## Visit spacing

Though part of the intervention package, the median time from visit date to next pharmacy appointment (visit space) between intervention and control sites were similar at 91 days (IQR:89–91) and 91 days (IQR: 63–91), respectively. During the study period the conventional refill was extended from 1-month to 3-months resulting is a mix of individuals in the control faculties receiving 1-month and 3-month refills. Despite the median similarity between intervention and control participants, there was a significant (t-test p-value < 0.001) difference in the mean visit space between intervention (mean: 86 days standard deviation [SD]: 18) and control (mean: 78 days SD: 24). Visit space was more variable for the control participants compared to intervention participants from visit to visit across the study window (Fig 6). There was a synergistic interaction effect observed between visit space and FT on participants arriving for a clinic visit ±3 days (Fig 5). The probability of an on-time return was highest among those at an intervention site receiving >85 day refill at 94% (95% CI: 93%-96%).

## Facility differences

The cumulative incidence of being greater than 28 days late for an appointment differed across facilities. The lowest incidence of being more than 28 days late was observed at an intervention

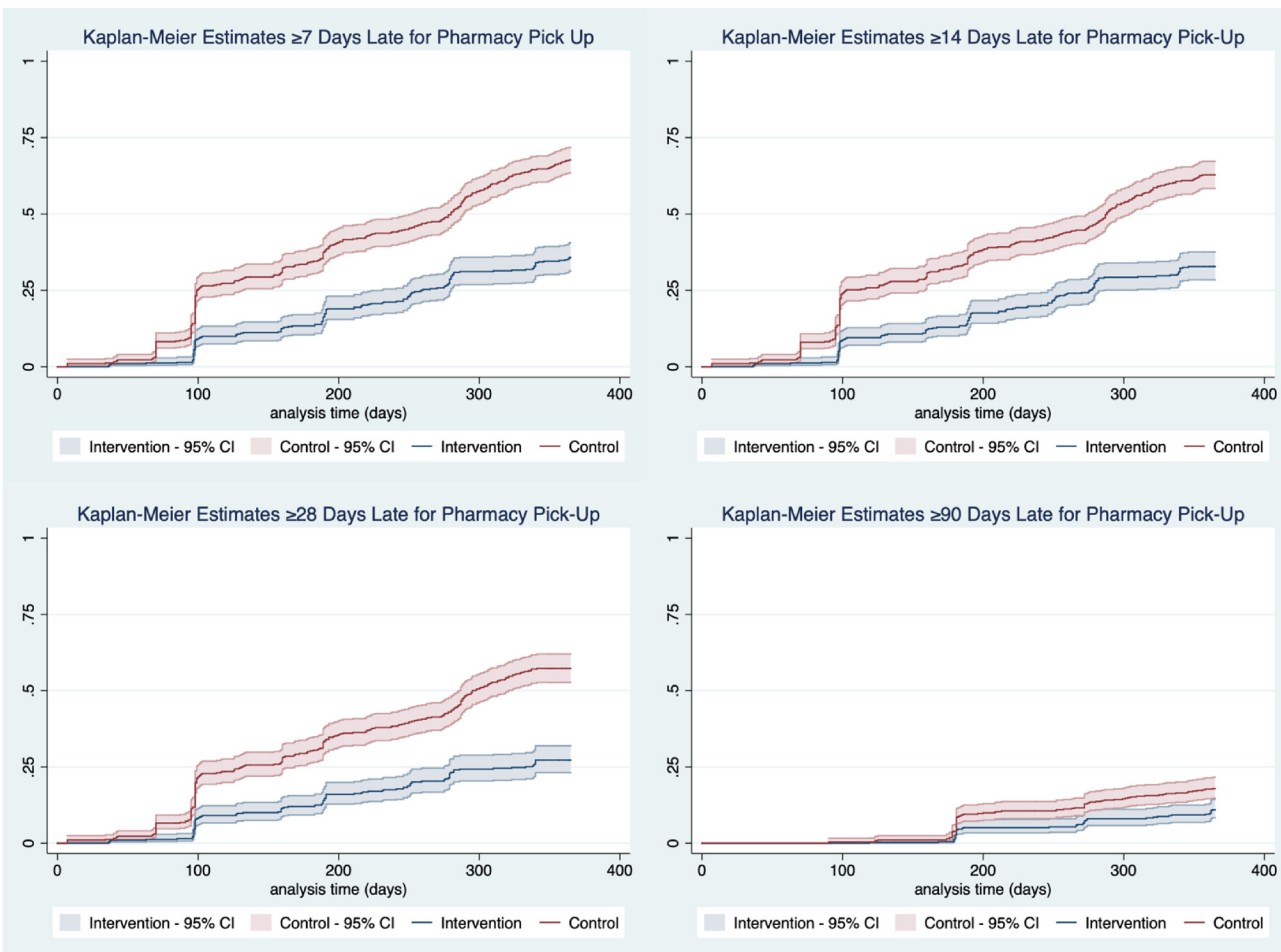

**Fig 3. Kaplan- Meier estimate curves for being late (>7 days, >14 days, >28 days, and >90 days) for pharmacy pick-up by intervention status with 95% confidence intervals.**

site (0.21, 95% CI: 0.16–0.26). Notably, two of the five control facilities had similar (i.e., not significantly different) incidence of late visit (>28 days) compared to the intervention facilities (0.33, 95% CI 0.25–0.43) (Table 3).

## Return after being late (>28 days)

Though not significant, cumulative incidence of return within 30 days after being late was higher for intervention participants 0.36 (95% CI 0.27–0.46) compared to the control 0.33 (95% CI 0.27–0.39) (Table 4). The median time to return was significantly longer for the control group compared to the intervention group at 51 days (IQR: 14–68 days) and 33 days (IQR: 3–63), respectively with a Wilcoxon rank-sum p-value = 0.01.

## Discussion

We found that the FT differentiated service delivery model significantly improved adherence to pharmacy pick-ups among persons living with HIV in Zambia. Intervention participants had lower cumulative incidence of late pharmacy pick-ups measured at 7, 14, and 28 days at one year of follow-up as well as with MPR. Although not significant, there was a difference

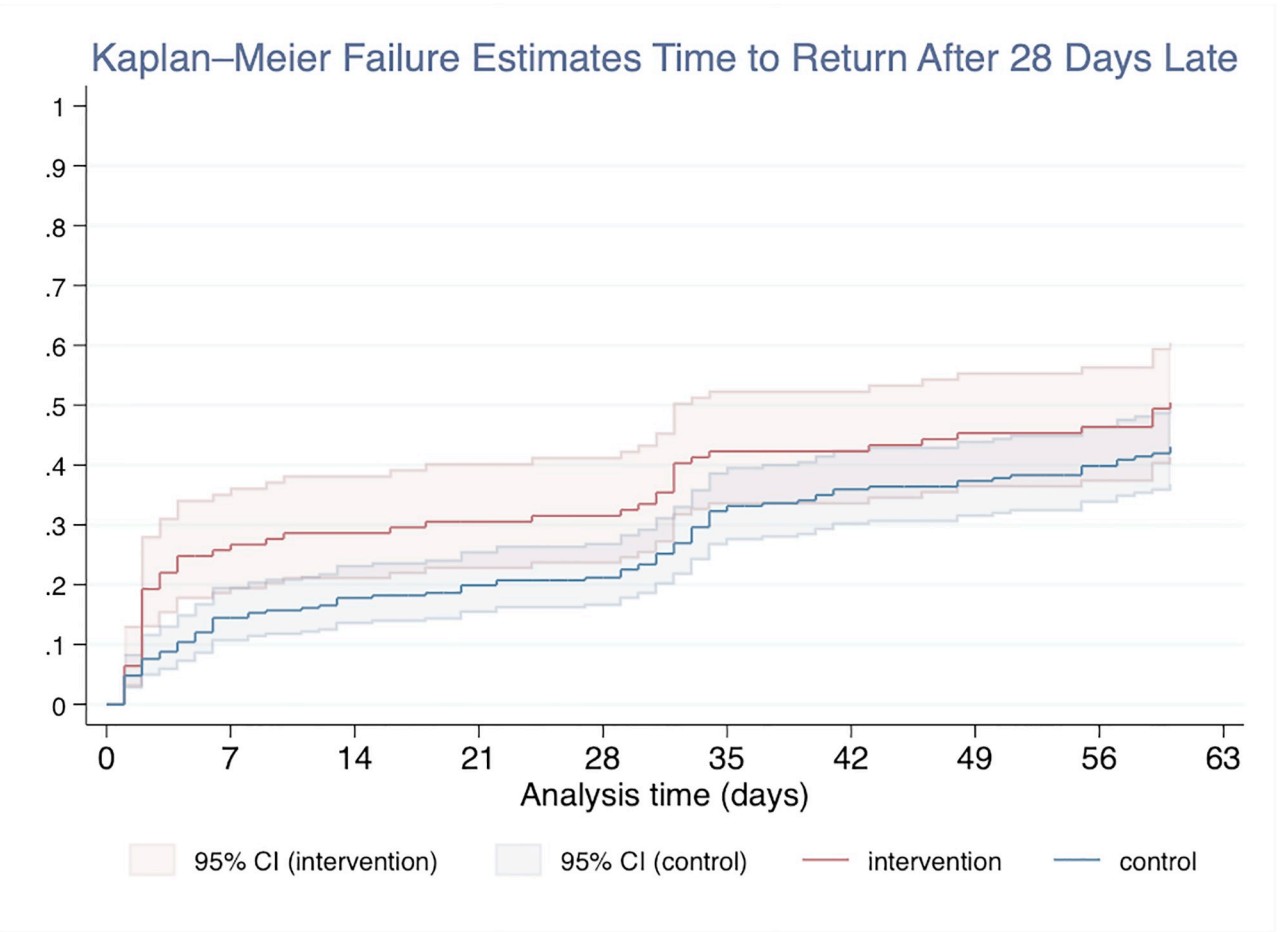

**Fig 4. Kaplan-Meier estimate curves for time to return after being more than 28 days late for ART refill by intervention status.**

between incidence of 90 days late for ART pick up compared to control at one year of follow-up. These effects were observed despite a key component of the intervention, refills of at least three months (multi-month scripting) amongst stable patients being adopted as national standard of HIV care during the study interval. Examination of the effect of FT across strata of prescription length revealed an interaction where longer duration and FT alone had little effect on late ART pick up whereas both together had far larger effects (Fig 5). Not only were intervention participants significantly less likely to be late for a drug pick, but they also returned to care more readily compared to those in the control group.

Because the FT model requires differential staffing and additional infrastructure, especially, a separate, clinical space dedicated to FT participant screening, program planners should carefully evaluate site suitability when implementing the FT model [13,14]. The expanding menu of DSD models being introduced in Zambia includes FT, urban adherence clubs/groups (groups consisting of approximately 30 meeting every two to three months during off-hours at an ART facility in an urban setting) and community adherence clubs/groups (groups consisting of 6 individuals meeting monthly for counseling, symptom check, and ART pick-up at a community site typically in a peri-urban or rural setting). Findings here as well as those published by others regarding alternate DSD models in sub-Saharan Africa, suggest that an array of differentiated care options may improve retention through reduced late ART pick up and lend to optimized HIV care in Zambia and beyond [13,15,16].

**Table 2. Mixed effects logistic regression analysis results for 90% medication possession ratio.**

| Variable | Level | Odds Ratio | p-value | 95% CI |
|---|---|---|---|---|
| Intervention | FT | 3.75 | 0.008 | 1.41–9.97 |
| Sex | Female | 1.00 (ref) | ref | ref |
| | Male | 1.12 | 0.606 | 0.73–1.71 |
| Age | 16–19 years | 1.00 (ref) | ref | ref |
| | 20–29 years | 0.79 | 0.477 | 0.41–1.51 |
| | 30–39 years | 0.83 | 0.598 | 0.42–1.64 |
| | 40+ years | 0.89 | 0.764 | 0.40–1.94 |
| Study CD4 | Cells/mm$^3$ | 1.00 | 0.353 | 1.00–1.00 |
| WHO Stage | 1 | 1.00 (ref) | ref | ref |
| | 2 | 0.96 | 0.886 | 0.57–1.62 |
| | 3 | 1.18 | 0.516 | 0.72–1.93 |
| | 4 | 0.98 | 0.969 | 0.33–2.91 |
| Years in HIV Care | | 0.98 | 0.542 | 0.92–1.05 |
| Marital Status | Married | 1.00 (ref) | ref | ref |
| | Single | 0.98 | 0.96 | 0.51–1.91 |
| | Divorced | 1.15 | 0.658 | 0.62–2.13 |
| | Widowed/er | 0.74 | 0.328 | 0.41–1.35 |
| Education Level | No Education | 1.00 (ref) | ref | ref |
| | < Grade 7 | 1.11 | 0.806 | 0.49–2.53 |
| | Grade 7–11 | 1.30 | 0.510 | 0.59–2.86 |
| | Grade 12 | 1.27 | 0.558 | 0.57–2.83 |
| | College/University | 1.19 | 0.804 | 0.30–4.69 |

Note: Adjusted for sex, age, CD4 cell count, WHO stage, years in HIV care, marital status, and education level; CI—confidence interval; ref—indicates variable referent group; WHO—World Health Organization.

Our findings are consistent with observations reported in rural Malawi where retention was significantly improved with modified visit intervals and alternate/expedited interim ART dispensation logistics compared to standard of care (aHR: 2.6, 95% CI: 2.2–3.1) [11]. However, we did not find a significant difference across age and gender as was observed in rural Malawi [11]. Age and gender associated differences in findings may be associated with the urban setting of our health care facilities coupled with limited twelve-month follow-up period given that retention has been documented as, generally, poorer among larger clinics often located in more urban settings in Sub Saharan-Africa set [17].

As ART programs continue to grow, and the burden on the health system continues to swell, identifying cost effective, sustainable solutions for decongesting clinics and improving efficiencies remains key. Additionally, ensuring that these DSD options are responsive and flexible to accommodate the necessary changes to meet patients where they are in their care journey including potential drift in and out of different models, changing social circumstances, stages of clinical well-being, and other individual needs. FT, whilst not able to address all barriers, is definitely a viable, cost-effective model that over the relative short term has shown to be successful at retaining patients and improving adherence.

## Limitations

There were several limitations in this study some of which are manifestations of the rapidly changing HIV care landscape in Zambia including programmatic shifts in ART refill intervals

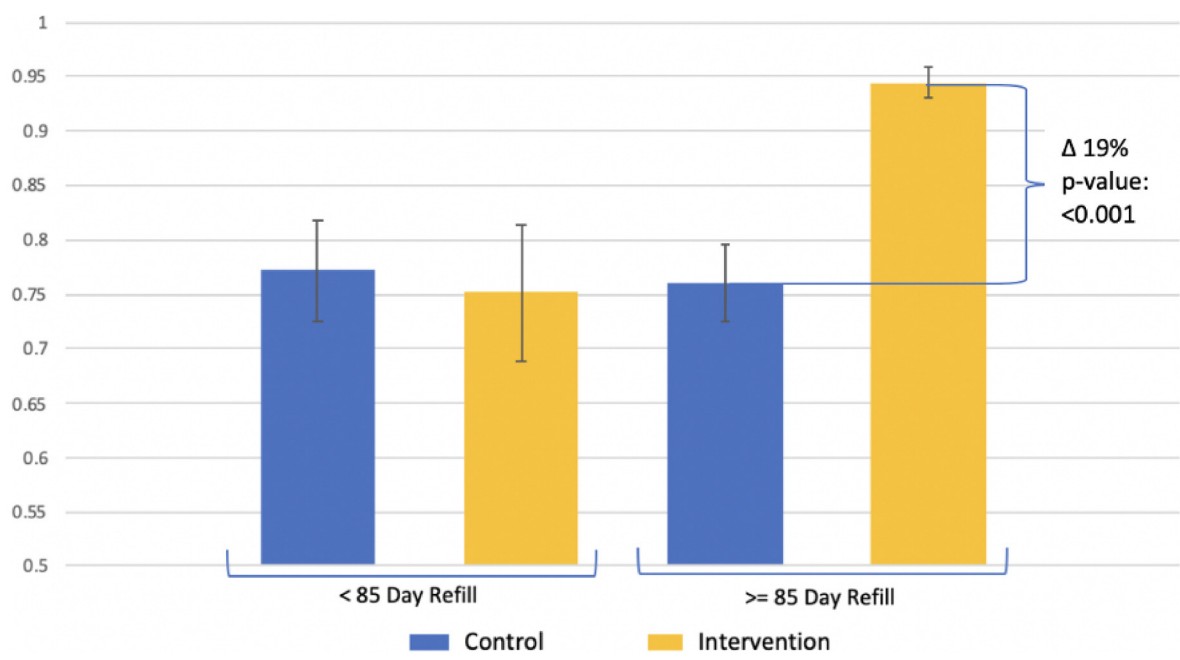

**Fig 5. Margins plot by ART refill (+/- 85 days) and intervention status.** Note: Model is adjusted for age, sex, WHO stage at HIV care initiation, time in HIV care, and CD4 cell count at HIV care initiation.

and increased in-care population related to out-patient department opt-out testing policy. Additionally, because we used routine data as transcribed into the electronic medical record to assess both intervention and control groups to assess late ART pick up it is possible that data entry errors occurred, though we do not expect these errors to differ by intervention status. Finally, we observed heterogeneity in the outcome of late ART pick up by facility, evidenced in Table 3, which may limit generalizability however, we describe underlying intervention and control participants and present adjusted analyses, with random effects at the facility level, to mitigate differences across facilities.

## Conclusions

Our study found that the FT differentiated service delivery model significantly improved adherence to pharmacy visits in Zambia. In addition, the apparently synergistic relationship between refill time and other elements of the FT model suggests that FT may further enhance the effects of extended visit spacing alone. Furthermore, the effect of FT on return to care seems positive. Although there were still a significant number of participants that were late for ART pick-up in the intervention group, they returned faster leading to higher MPR and reduced likelihood of risk for resistance.

## Implications

The FT model is effective in improving retention in Zambia and should be considered a viable model to optimize retention in Zambia and, possibly beyond. The FT model may also be implemented to maximize medication possession ratio thereby reducing the risk of virologic failure and subsequent transmission. Previous research shows that extended refill interval is

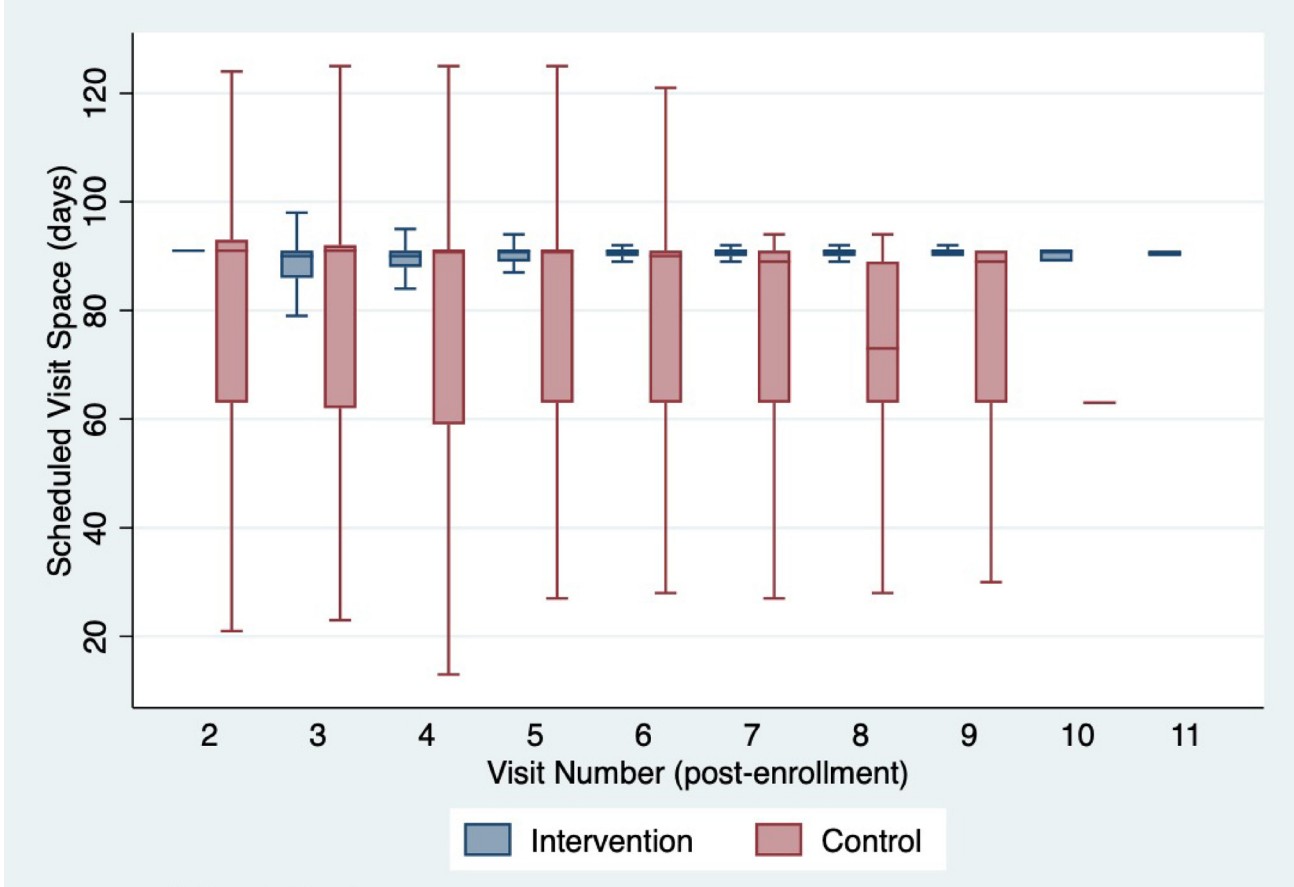

**Fig 6. Box plot of time between visit appointments.** Note: Boxes indicate median (line in box) and interquartile range (ends of box) whiskers correspond to 95% CI.

associated with improved retention however, we found that the effect of the intervention as a package went beyond the improvement observed in implementation of extended visit spacing alone. Qualitative data suggest that intervention participants may have been more likely to visit the clinic on time due to a perceived increased accountability as a formally enrolled FT participant. This size of this effect has not been measured but it may be challenging to maintain this effect at scale (i.e. all eligible stable patients are offered FT participation over an extended period at a given clinic).

**Table 3. Cumulative incidence of visit >28 days after appointment by facility at one year of follow-up.**

| Clinic Name | Status | Cumulative Incidence | 95% CI |
|---|---|---|---|
| I1 | FT | 0.35 | 0.28–0.43 |
| I2 | FT | 0.21 | 0.16–0.26 |
| C1 | Control | 0.33 | 0.25–0.43 |
| C2 | Control | 0.73 | 0.64–0.81 |
| C3 | Control | 0.70 | 0.60–0.79 |
| C4 | Control | 0.36 | 0.26–0.50 |
| C5 | Control | 0.51 | 0.42–0.61 |

**Table 4. Cox hazards model for >28 days late for appointment.**

| Variable | Level | aHR | p-value | 95% CI |
|---|---|---|---|---|
| Intervention Status | Control | 2.58 | <0.001 | 1.55–4.31 |
| | Intervention | | | |
| Sex | Female | 1.00 (ref) | ref | ref |
| | Male | 0.97 | 0.828 | 0.71–1.31 |
| Age Category | 16–24 years | 1.62 | 0.116 | 0.89–2.95 |
| | 24–34 years | 1.28 | 0.280 | 0.82–2.01 |
| | 35–44 years | 1.40 | 0.138 | 0.900–2.17 |
| | 45–54 years | 1.50 | 0.009 | 1.11–2.01 |
| | 55+ years | 1.00 (ref) | ref | ref |
| Baseline CD4 | Cells/mm$^3$ | 1.00 | 0.951 | 1.00–1.00 |
| WHO Stage | 1 | 1.00 (ref) | ref | ref |
| | 2 | 1.10 | 0.587 | 0.78–1.54 |
| | 3 | 1.12 | 0.396 | 0.86–0.86 |
| | 4 | 0.94 | 0.853 | 0.46–1.90 |
| Time in HIV Care | years | 1.00 | 0.696 | 1.00–1.00 |
| Marital Status | Married | 1.00 (ref) | ref | ref |
| | Single | 0.99 | 0.956 | 0.77–1.28 |
| | Divorced | 0.97 | 0.836 | 0.691.35 |
| | Widowed | 1.13 | 0.392 | 0.85–1.51 |
| Educational Level | No formal education | 1.00 (ref) | ref | ref |
| | Grade 1–6 | 1.05 | 0.861 | 0.59–1.88 |
| | Grade 7–11 | 1.25 | 0.458 | 0.69–2.24 |
| | Grade 12 | 0.99 | 0.983 | 0.58–1.72 |
| | College/University | 1.27 | 0.477 | 0.65–2.48 |

Note: aHR—adjusted hazard ratio; 95% CI—95% confidence interval; ref—indicates variable referent group.

## Supporting information

**S1 Fig. Kaplan-Meier estimates (>7 days late) among A) control participants and B) intervention participants by antiretroviral therapy refill spacing with risk tables.**
(EPS)

## Acknowledgments

The authors would like to thank the CIDRZ study team, the study participants, the study clinic staff, and the Zambian Ministry of Health for their contributions to the study.

## Author Contributions

**Conceptualization:** Carolyn Bolton Moore, Monika Roy, Hojoon Sohn, David W. Dowdy, Nancy Padian, Charles B. Holmes, Elvin H. Geng, Izukanji Sikazwe.

**Data curation:** Jake M. Pry, Mpande Mukumbwa-Mwenechanya, Stephanie Topp, Monika Roy, Charles B. Holmes, Elvin H. Geng.

**Formal analysis:** Carolyn Bolton Moore, Jake M. Pry, Ingrid Eshun-Wilson, Chanda Mwamba, Monika Roy, David W. Dowdy, Elvin H. Geng.

**Funding acquisition:** Carolyn Bolton Moore, Monika Roy, Nancy Padian, Charles B. Holmes, Elvin H. Geng, Izukanji Sikazwe.

**Investigation:** Carolyn Bolton Moore, Jake M. Pry, Ingrid Eshun-Wilson, Hojoon Sohn, David W. Dowdy, Nancy Padian, Charles B. Holmes, Elvin H. Geng, Izukanji Sikazwe.

**Methodology:** Carolyn Bolton Moore, Jake M. Pry, Ingrid Eshun-Wilson, Stephanie Topp, Chanda Mwamba, Monika Roy, Hojoon Sohn, David W. Dowdy, Charles B. Holmes, Elvin H. Geng.

**Project administration:** Carolyn Bolton Moore, Mpande Mukumbwa-Mwenechanya, Charles B. Holmes, Izukanji Sikazwe.

**Resources:** Carolyn Bolton Moore, Jake M. Pry, Charles B. Holmes, Elvin H. Geng.

**Software:** Jake M. Pry.

**Supervision:** Carolyn Bolton Moore, Mpande Mukumbwa-Mwenechanya.

**Validation:** Jake M. Pry, Ingrid Eshun-Wilson, Stephanie Topp, Chanda Mwamba, Monika Roy, Hojoon Sohn, David W. Dowdy, Charles B. Holmes.

**Visualization:** Jake M. Pry.

**Writing – original draft:** Carolyn Bolton Moore, Jake M. Pry, Ingrid Eshun-Wilson, Stephanie Topp, Monika Roy, Hojoon Sohn, David W. Dowdy, Nancy Padian, Charles B. Holmes, Elvin H. Geng, Izukanji Sikazwe.

**Writing – review & editing:** Carolyn Bolton Moore, Jake M. Pry, Charles B. Holmes, Elvin H. Geng.

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
