## [Decision Letter · Decision Letter 0]

23 Sep 2021

PGPH-D-21-00138

Effects of a Fast Track (FT) service delivery model among stable HIV patients in Lusaka Zambia

Dear Dr. Pry,

Thank you for submitting your manuscript to PLOS Global Public Health. After careful consideration, we feel that it has merit but does not fully meet PLOS Global Public Health’s publication criteria as it currently stands. Therefore, we invite you to submit a revised version of the manuscript that addresses the points raised during the review process.

We look forward to receiving your revised manuscript.

Kind regards,

Joel Msafiri Francis, MD, MS, PhD

Academic Editor

Journal Requirements:

1. Please include a copy of the interview guide used in the study, in both the original language and English, as Supporting Information, or include a citation if it has been published previously.

2. During our internal checks, the in-house editorial staff noted that you conducted research or obtained samples in another country. Please check the relevant national regulations and laws applying to foreign researchers and state whether you obtained the required permits and approvals. Please address this in your ethics statement in both the manuscript and submission information. In addition, please ensure that you have suitably acknowledged the contributions of any local collaborators involved in this work in your authorship list and/or Acknowledgements. Authorship criteria is based on the International Committee of Medical Journal Editors (ICMJE) Uniform Requirements for Manuscripts Submitted to Biomedical Journals.

3. You indicated that you had ethical approval for your study. In your Methods section, please ensure you have also stated whether you obtained consent from parents or guardians of the minors included in the study or whether the research ethics committee or IRB specifically waived the need for their consent.

4. Please provide separate figure files in .tif or .eps format only, and remove any figures embedded in your manuscript file.  If you are using LaTeX, you do not need to remove embedded figures.

5. Please update the completed 'Competing Interests' statement, including any COIs declared by your co-authors. If you have no competing interests to declare, please state "The authors have declared that no competing interests exist". Otherwise please declare all competing interests beginning with the statement "I have read the journal's policy and the authors of this manuscript have the following competing interests:"

6. In the online submission form, you indicated that "De-identified data will be made available upon request".

7. Please amend your] detailed Financial Disclosure statement. This is published with the article, therefore should be completed in full sentences and contain the exact wording you wish to be published.

i) Please include all sources of funding (financial or material support) for your study. List the grants (with grant number) or organizations (with url) that supported your study, including funding received from your institution. 

ii). State the initials, alongside each funding source, of each author to receive each grant.

iii). State what role the funders took in the study. If the funders had no role in your study, please state: “The funders had no role in study design, data collection and analysis, decision to publish, or preparation of the manuscript.”

iv). If any authors received a salary from any of your funders, please state which authors and which funders.

8. In your Financial Disclosure, you indicated "No financial disclosures to report", but in the Funding Information you indicated the following:

Bill and Melinda Gates Institute for Population and Reproductive Health, OPP1105071, Dr. Charles B Holmes

National Institutes of Health, AI134413, Dr. Elvin H Geng

Please revise the Funding Information field to reflect funding received.

Reviewers' comments:

Reviewer's Responses to Questions

**Comments to the Author**

1. Does this manuscript meet PLOS Global Public Health’s publication criteria? Is the manuscript technically sound, and do the data support the conclusions? The manuscript must describe methodologically and ethically rigorous research with conclusions that are appropriately drawn based on the data presented.

Reviewer #1: Partly

Reviewer #2: Yes

2. Has the statistical analysis been performed appropriately and rigorously?

Reviewer #1: Yes

Reviewer #2: Yes

3. Have the authors made all data underlying the findings in their manuscript fully available (please refer to the Data Availability Statement at the start of the manuscript PDF file)?

Reviewer #1: No

Reviewer #2: No

4. Is the manuscript presented in an intelligible fashion and written in standard English?

Reviewer #1: Yes

Reviewer #2: Yes

5. Review Comments to the Author

Reviewer #1: Many thanks for this interesting paper on fast track ART delivery. The area of how to improve patient retention and viral suppression with DSD models is very critical. I do however have some major concerns with the paper as it is currently written

1. The methodology is extremely unclear. The abstract describes this as a controlled study whereas the methods section in the main paper presents this as part of a cluster-randomized trial NCT02776254. While this trial does mention fast track in the description on clinical trials, what is being described here is not cluster randomized as far as I can see. The description in the methods, states that sites were purposively sampled with the “best possible counterfactuals”- however it is unclear how the sites were actually sampled and why only 2 intervention sites were selected and 5 control sites? In the description of the larger study it states that from 28 sites 13 were selected to be intervention sites (It seems that it should be 14, and note that the primary paper from the trial states that 10 were intervention and control sites?). It then suggests that the 2 FT intervention sites and the 5 FT control sites were selected from the intervention sites. Is this correct? If so they presumably had other DSD interventions that are not mentioned here- could these have biased the outcomes of this comparison?

2. Although unclear whether this is a cluster-randomized study or not the “participants” in this study are actually the clinics as it is these that were allocated into FT intervention or control. As such the flow diagram shown as figure 2 does not match this allocation schema. Furthermore, in the results section you name the clinics. As participants would never be named it is generally not acceptable to name clusters and I would suggest they should only be identified by a study number. However, since they are named they do not seem to be “best possible counterfactuals”. I am concerned around the potential biases from extremely different clinics in the intervention and control arms. I cannot see any data presented to demonstrate that the clinics in the 2 arms were similar with respect to numbers of patients in care, numbers of staff, facilities, waiting times, retention rates per-intervention etc. I think this needs to be provided.

3. The baseline characteristics of the patients in the intervention and control groups differed in several characteristics which seems to imply that if they were truly randomly sampled then the clinic populations differed, again raising questions of recruitment bias.

4. Kaplan Meier curves should include the numbers of participants in the respective groups at each timepoint

5. Figure 5 is unclear. In the text it states that this is demonstrating MPR, however the legend on this plot is “Probability of making a visit (3 days) by intervention and refill interval”. This legend does not make sense grammatically or practically. Please consider amending the title and provide explanation of what this graphic shows.

6. Results visit spacing is again unclear. I think you are saying that the appointments given were similar- median 91 days – although it seems the control group had quite a different IQR? Can you explain this given that all patients had a 90 day supply? Or is this to do with the introduction of MMD (see later question)? The next sentence states there was a significant difference in the “median time”. I don’t understand this- do you mean the actual time between visits or the appointment? Please make this clearer.

7. The rest of this section is also unclear and seems to refer back to figure 5 which I think relates to this rather than the section where it is placed. Please consider rewriting this section to clarify what exactly is meant.

8. The study is described as “mixed methods” and while the team report on qualitative data collection including FGD and IDI they provide no information in the methods about analysis of qualitative data. Qualitative results are not presented as might be expected from a thorough analysis and so it is questionable what this adds to the paper. Whilst I am highly in favour of mixed methods research, I would strongly suggest that either more space is given to describe the analytical process and to present the results fully or to split the quantitative and qualitative methods and write each up more fully.

9. Are there any data available about time spent in the clinic, time of the staff to deliver the model, costs from the provider and patient perspective? These findings would strengthen the manuscript greatly.

10. In the discussion the authors bring up multi-month scripting for the first time. If a change occurred mid-way through the intervention then this should be mentioned in the methods and possibly a timeline provided or some consideration of a sensitivity analysis pre and post changes. Was the MMD for longer than the 3 month dispensing being described in the methods?

11. Limitations- the greatest limitations are the undoubtable biases that will occur since this was not a randomised study. This should be discussed.

12. Implications. I am not clear how the authors can state that the effects of FT went beyond those of MMD. As far as I can see these were not directly compared in this study. Suggest removing or revising

13. Data availability- whilst I understand the sensitivities of this clinical data being available I think that additional data should be provided in the manuscript- such as the numbers requested to accompany Kaplan Meier curves and showing more of the actual numbers in tables.

Minor issues

1. Discussion- unfortunately lines are not numbered- paragraph 5 incomplete sentence “……95:95:95 targets is n sustaining….”

2. Same para “FT….not being an elixir for all” this is strange phrasing consider rewriting

Reviewer #2: It should be noted that the data is not publicly available without permission due to being MoH data and this is an acceptable exception.

The statistical rigor is exceptional for this manuscript - the qualitative methods need to be better described, particularly around coding, thematic analysis, and rigor of results.

There is also a need to straightforwardly convey the quantitative results - I found it difficult to parse the key results with the acronyms and heavily described statistics. Best to lead with lay description and back-up with relevant statistics. I would not call it jargon, but felt a person needed to be fairly expert with these indicators to follow the thread.

This manuscript also fills a critical gap in the response and evidence on DSD and retention in care. Some minor edits would help to convey the context, standard of care in Zambia and how this changed over study duration, resource limitations of the intervention which restricted it to two intervention sites (infrastructure - i.e., space, which comes in late in the MS).

As far as qualitative results, it would also be helpful to describe why the pre-implementation FG findings are described, and also if they informed the intervention at all. The results write-up would benefit from presenting the qual findings longitudinally. The interspersion of quotes is especially effective in the later results paragraph, and would be good to interweave in the beginning qual paragraphs.

This is an excellent study and definitely fills a needed evidence gap in the HIV response, and casts a light on the difficulty of conducting implementation studies in vivo, in a dynamic policy and health system landscape.

6. PLOS authors have the option to publish the peer review history of their article (what does this mean?). If published, this will include your full peer review and any attached files.

**Do you want your identity to be public for this peer review?** For information about this choice, including consent withdrawal, please see our Privacy Policy.

Reviewer #1: No

Reviewer #2: No

---

## [Decision Letter · Decision Letter 1]

6 Feb 2022

PGPH-D-21-00138R1

Effects of a Fast Track (FT) service delivery model among stable HIV patients in Lusaka Zambia

Dear Dr. Pry,

Thank you for submitting your manuscript to PLOS Global Public Health. After careful consideration, we feel that it has merit but does not fully meet PLOS Global Public Health’s publication criteria as it currently stands. Therefore, we invite you to submit a revised version of the manuscript that addresses the points raised during the review process.

We look forward to receiving your revised manuscript.

Kind regards,

Joel Msafiri Francis, MD, MS, PhD

Academic Editor

Journal Requirements:

1. Please ensure that you refer to Figure 2 in your text as, if accepted, production will need this reference to link the reader to the figure.

Additional Editor Comments (if provided):

Reviewers' comments:

Reviewer's Responses to Questions

**Comments to the Author**

1. If the authors have adequately addressed your comments raised in a previous round of review and you feel that this manuscript is now acceptable for publication, you may indicate that here to bypass the “Comments to the Author” section, enter your conflict of interest statement in the “Confidential to Editor” section, and submit your "Accept" recommendation.

Reviewer #1: (No Response)

Reviewer #3: (No Response)

2. Does this manuscript meet PLOS Global Public Health’s publication criteria? Is the manuscript technically sound, and do the data support the conclusions? The manuscript must describe methodologically and ethically rigorous research with conclusions that are appropriately drawn based on the data presented.

Reviewer #1: Partly

Reviewer #3: Partly

3. Has the statistical analysis been performed appropriately and rigorously?

Reviewer #1: No

Reviewer #3: I don't know

4. Have the authors made all data underlying the findings in their manuscript fully available (please refer to the Data Availability Statement at the start of the manuscript PDF file)?

Reviewer #1: No

Reviewer #3: No

5. Is the manuscript presented in an intelligible fashion and written in standard English?

Reviewer #1: No

Reviewer #3: No

6. Review Comments to the Author

Reviewer #1: I am afraid that the revisions do not seem to be complete or to address all of the points yet.

I am happy about the data not being available due to this being MOH data and so this is not an issue.

My major issue is still with the design and the assumptions that the authors make regarding the comparability of the clinics in the two groups. Whilst the design has become clearer- that from a pool of 28 clinics 16 were randomly selected for DSD interventions of which 2 were selected to have FT ( the subject of this paper). From what I can understand is that of 16 with DSD interventions 5 were selected to be FT controls ( but were therefore not full SoC clinics as they had some DSD?).

Since this does appear to be a cluster randomised trial its reporting does not follow the standard consort guidelines in that the abstract and title do not include information that this is a randomised trial, we do not have information about the randomisation process nor blinding etc. A consort checklist should be provided. Since the number of clusters is so small and uneven ( there really need to be at least 6 per arm to have any confidence in this) there are likely to be many existing biases that would not be overcome by randomisation and therefore assumptions about these are incorrect. This needs to be discussed more in the discussion -

The clinics are still named- which contravenes their confidentiality as research participants

I note that the authors have chosen to remove the qualitative section to another paper but the abstract and methods still imply that this is a mixed methods study and so needs further editing ( methods line 173/4, abstract needs revision throughout)

The abstract does not now fit the details in the main body of the article and needs a thorough revision to match the methods etc

Reviewer #3: Thank you for the opportunity to review this manuscript, which addresses an important question regarding the impact of the Fast Track differentiated HIV service delivery models on outcomes such as missed visits and retention in care. While the topic is of high interest, especially the deep dive into programmatic data, the manuscript itself is not ready for publication. In particular, removing the qualitative data from the methods and results sections, but leaving them in the abstract, introduction and discussion sections is problematic. A more minor point is that manuscript needs a close edit – there are a few typos, grammatical errors, and misspellings. More detailed notes are attached.

7. PLOS authors have the option to publish the peer review history of their article (what does this mean?). If published, this will include your full peer review and any attached files.

**Do you want your identity to be public for this peer review?** For information about this choice, including consent withdrawal, please see our Privacy Policy.

Reviewer #1: No

Reviewer #3: No

---

## [Decision Letter · Decision Letter 2]

28 Apr 2022

PGPH-D-21-00138R2

A controlled study to assess the effect of a Fast Track (FT) service delivery model among stable HIV patient in Lusaka, Zambia

Dear Dr. Pry,

Thank you for submitting your manuscript to PLOS Global Public Health. After careful consideration, we feel that it has merit but does not fully meet PLOS Global Public Health’s publication criteria as it currently stands. Therefore, we invite you to submit a revised version of the manuscript that addresses the points raised during the review process.

**Specifically, It would be more helpful to address all the reviewer’s comments. There is an overall lack of responsiveness to the comments made by reviewers. It would be more helpful to structure the methods section properly and to be transparent in the methods used.**

We look forward to receiving your revised manuscript.

Kind regards,

Joel Msafiri Francis, MD, MS, PhD

Academic Editor

Journal Requirements:

Additional Editor Comments (if provided):

**It would be more helpful to address all the reviewer’s comments. There is an overall lack of responsiveness to the comments made by reviewers. It would be more helpful to structure the methods section properly and to be transparent in the methods used.**

Reviewers' comments:

Reviewer's Responses to Questions

**Comments to the Author**

1. If the authors have adequately addressed your comments raised in a previous round of review and you feel that this manuscript is now acceptable for publication, you may indicate that here to bypass the “Comments to the Author” section, enter your conflict of interest statement in the “Confidential to Editor” section, and submit your "Accept" recommendation.

Reviewer #1: (No Response)

2. Does this manuscript meet PLOS Global Public Health’s publication criteria? Is the manuscript technically sound, and do the data support the conclusions? The manuscript must describe methodologically and ethically rigorous research with conclusions that are appropriately drawn based on the data presented.

Reviewer #1: Partly

3. Has the statistical analysis been performed appropriately and rigorously?

Reviewer #1: Yes

4. Have the authors made all data underlying the findings in their manuscript fully available (please refer to the Data Availability Statement at the start of the manuscript PDF file)?

Reviewer #1: No

5. Is the manuscript presented in an intelligible fashion and written in standard English?

Reviewer #1: Yes

6. Review Comments to the Author

Reviewer #1: Please see attachment

7. PLOS authors have the option to publish the peer review history of their article (what does this mean?). If published, this will include your full peer review and any attached files.

**Do you want your identity to be public for this peer review?** For information about this choice, including consent withdrawal, please see our Privacy Policy.

Reviewer #1: No

---

## [Editor Report · Decision Letter 3]

13 Jul 2022

A controlled study to assess the effect of a Fast Track (FT) service delivery model among stable HIV patient in Lusaka, Zambia

PGPH-D-21-00138R3

Dear Dr. Pry,

We are pleased to inform you that your manuscript 'A controlled study to assess the effect of a Fast Track (FT) service delivery model among stable HIV patient in Lusaka, Zambia' has been provisionally accepted for publication in PLOS Global Public Health.

Best regards,

Joel Msafiri Francis, MD, MS, PhD

Academic Editor
